# Acute Kidney Injury in Liver Cirrhosis

**DOI:** 10.3390/diagnostics13142361

**Published:** 2023-07-13

**Authors:** Rose Mary Attieh, Hani M. Wadei

**Affiliations:** Department of Transplant, Division of Kidney and Pancreas Transplant, Mayo Clinic, Jacksonville, FL 32224, USA; attieh.rosemary@mayo.edu

**Keywords:** hepatorenal syndrome

## Abstract

Acute kidney injury (AKI) is common in cirrhotic patients affecting almost 20% of these patients. While multiple etiologies can lead to AKI, pre-renal azotemia seems to be the most common cause of AKI. Irrespective of the cause, AKI is associated with worse survival with the poorest outcomes observed in those with hepatorenal syndrome (HRS) and acute tubular necrosis (ATN). In recent years, new definitions, and classifications of AKI in cirrhosis have emerged. More knowledge has also become available regarding the benefits and drawbacks of albumin and terlipressin use in these patients. Diagnostic tools such as urinary biomarkers and point-of-care ultrasound (POCUS) became available and they will be used in the near future to differentiate between different causes of AKI and direct management of AKI in these patients. In this update, we will review these new classifications, treatment recommendations, and diagnostic tools for AKI in cirrhotic patients.

## 1. Introduction

Renal dysfunction poses a heavy burden to patients with liver disease. Acute kidney injury (AKI) affects 30–50% of hospitalized patients with liver cirrhosis [1,2,3] and leads to the development of acute kidney disease (AKD) and de novo chronic kidney disease (CKD). AKI is also associated with a myriad of complications in cirrhotic patients including prolonged hospitalization, and decreased survival [1,4,5,6,7]. Novel tools are becoming available to assist in diagnosing and treating this large population of patients. In the work herein, we aim to review the definition, staging, etiology, and epidemiology of AKI in cirrhotic patients. We also discuss the pathophysiology of hepatorenal syndrome (HRS) and present updates in its diagnosis and management strategies.

## 2. Definitions

Prior to 2015, an arbitrary serum creatinine (SCr) cut-off of ≥1.5 mg/dL was used to diagnose renal dysfunction, including AKI, in cirrhotic patients. The lack of unified diagnostic criteria for AKI led to wide variability in the reported incidence of AKI in this population, which varied from 20% to 50% [8,9]. In 2015, the International Club of Ascites (ICA) revised the definition criteria for AKI in cirrhosis and removed the traditional requirement of a SCr ≥1.5 mg/dL (133 µmol/L) [10]. The new definition, combining Acute Kidney Injury Network (AKIN), Risk Injury Failure Loss of Renal function (RIFLE), and End-Stage Renal Disease criteria, comprises 3 stages: stage 1, increase in SCr ≥0.3 mg/dL (26.5 μmol/L) within 48 h or increase ≥1.5–1.9-fold from baseline; stage 2, increase in SCr 2–2.9-fold from baseline; and stage 3, increase in SCr ≥3-fold from baseline or SCr ≥4.0 mg/dL (353.6 μmol/L), with an acute increase of ≥0.3 mg/dL (26.5 μmol/L) or initiation of renal-replacement therapy [11].

The HRS diagnostic criteria have also been recently redefined to reflect that HRS is a spectrum of diseases rather than a single entity (Table 1). The terms HRS-1 and HRS-2 were replaced by HRS-AKI and HRS-NAKI (non-AKI), respectively. HRS-AKI refers to an increase in SCr of ≥0.3 mg/dL within 48 h, or an increase in SCr ≥50% using the last available value of outpatient SCr within 3 months as the baseline value. HRS-AKI can only be diagnosed in a patient with decompensated cirrhosis and AKI without improvement in kidney function after diuretic withdrawal and plasma volume expansion with albumin infusion (1 g/kg body weight per day) for 2 days. It also necessitates the absence of shock, nephrotoxic drug exposure, or structural kidney disease (indicated by proteinuria >500 mg per day, microhematuria >50 red blood cells per high-power field, and/or abnormal renal ultrasonography). On the other hand, HRS-NAKI comprises both HRS-AKD and HRS-CKD. HRS-AKD is defined by an estimated glomerular filtration rate (eGFR) of <60 mL/min per 1.73 m^2^ for <3 months that is not otherwise explained by another pathological process, and HRS-CKD is defined by an eGFR of <60 mL/min per 1.73 m^2^ for ≥3 months in the absence of structural causes [12].

The HRS-AKI classification follows the same staging system proposed by the ICA but divides stage 1 AKI into 1a: SCr <1.5 mg/dL and 1b: SCr ≥1.5 mg/dL. Multiple studies have shown that this revised HRS-AKI staging system closely correlates with patient survival and is an excellent prognostic tool. For example, Huelin et al. found that the HR for 90-day mortality was 1.19 for stage 1A (95% CI 0.53–2.69), 2.54 for stage 1B (95% CI 1.45–4.44), 2.36 for stage 2 (95% CI 1.21–4.60), and 2.58 for stage 3 (95% CI 1.29–5.17) [2]. It is important to note however that a reduction in urine output is still not included in the current HRS-AKI classification despite its known diagnostic and prognostic significance. In fact, incorporating oliguria into the diagnostic criteria for HRS-AKI increased AKI incidence from 57.9% to 82.5% in one study. In addition, patients with stage 3 AKI defined by urine output criteria were found to have the highest hospital mortality regardless of AKI staging by standard criteria [13].

It is important to mention that all definitions of AKI in cirrhotic patients still use SCr despite its multiple limitations in liver cirrhosis [14]. Indeed, one of the major drawbacks of SCr is the time lag between the onset of kidney injury and the rise in SCr levels [15]. This period may potentially represent an important therapeutic window when the pathophysiological driver of HRS can still be reversed. Another important limitation is that SCr lacks discriminatory ability with regards to the etiology of kidney injury. In addition, patients with liver disease have sarcopenia, decreased hepatic synthesis of creatinine, and increased tubular secretion of creatinine which translates into overestimation of the actual GFR [16,17,18]. Furthermore, ascites increases the total volume of distribution of creatinine, leading to an artificially reduced serum concentration [19]. Whether other biomarkers of the GFR such as cystatin C will allow early diagnosis and management of AKI in this population remains to be studied.

## 3. Differential Diagnoses for AKI in Cirrhosis

Hepatorenal syndrome classically occurs in individuals with ascites, hypotension, oliguria, bland urinary sediment, and absence of proteinuria. However, before making such a diagnosis with certainty, one must judiciously look for the numerous competing causes of AKI [20].

Table 2 summarizes differential diagnoses of kidney injury in cirrhotic patients and their potential causes.

## 4. Epidemiology and Prognostic Implications

A recent multi-center case series from the US examined the epidemiology of AKI in the modern era using the 2015 IAC criteria. The cause of AKI was determined by retrospective chart review and confirmed by a second adjudicator (and third if needed). Out of 2063 patients with AKI and cirrhosis, 44.3% had pre-renal AKI and 30.4% had ATN. HRS-AKI was deemed to be the culprit in only 12.1% of cases. Pre-renal AKI had the best prognosis with 22% 90-day mortality, whereas HRS and ATN had 49% and 53% 90-day mortality, respectively (manuscript currently under review). These findings indicate that HRS in the modern era is no longer associated with the worst prognosis as it has been reported historically, and it now confers the same mortality risk as ATN. For example, in a prospective study of 562 patients with renal failure and cirrhosis (91% of these patients had AKI) published in 2011, the OR for 3-months mortality was 3.48 in HRS (95% CI 1.48–8.17, *p* = 0.004) and 2.62 in pre-renal AKI (95% CI 1.15–5.98, *p* = 0.022) [21]. Similarly, in 2013, Fargundes et al. found that HRS had the least survival probability in cirrhotic patients, followed by infection-related AKI, hypovolemia, and nephrotoxicity, respectively [22]. Perhaps this change in mortality risk was driven by the advent of new effective therapies for HRS in the past decade.

## 5. Differentiating Different Causes of AKI in Cirrhosis

### 5.1. Clinical Assessment

Physical examinations have multiple shortcomings in the context of cirrhosis, and discerning intra- from extra-vascular volume can be quite challenging. Although ascites and pedal edema are indicative of overall salt and water retention and are markers of an overall increase in total body extracellular volume, they can also be present in individuals with effective arterial blood volume (EABV) contraction. Lack of peripheral edema, however, is indicative of a decrease in EABV and is suggestive of pre-renal azotemia. The presence of ascites is a pre-requisite to establish a diagnosis of HRS.

### 5.2. Conventional Diagnostic Tools

Urine analysis, microscopic examination of the urinary sediment, and urine electrolytes are typically obtained at time of SCr elevation as well as a renal ultrasound to rule out obstructive uropathy. Low urinary fractional sodium excretion (FeNa) (<1%) is one the hallmark laboratory findings in HRS, but can also be observed in multiple other causes of kidney injury such as pre-renal azotemia, ATN, and MPGN (sensitivity 100% and specificity 14% for HRS) [23]. Conversely, in patients with HRS who are on diuretics, a urinary sodium excretion of >10 mEq/L can be deceiving. The fractional excretion of urea (FeURea) is not affected by diuretic use and is also proposed as a useful tool for differentiating ATN from non-ATN etiologies with high diagnostic accuracy [24].

Table 3 summarizes key diagnostic features of pre-renal azotemia (PRA), ATN and HRS. 

### 5.3. Novel Biomarkers

The ideal biomarker for renal function in cirrhosis should help predict AKI development, define the etiology of AKI, and predict AKI progression and outcomes [25]. Several novel biomarkers have been studied, such as Cystatin C, neutrophil gelatinase-associated lipocalin (NGAL), kidney injury molecule-1 (KIM1), liver fatty acid-binding protein (L-FABP), and IL-18, but their use is still limited. The timing of checking the biomarker relative to the AKI also affects the diagnostic accuracy.

Table 4 discusses the most important novel biomarkers as well as their utility and pitfalls. 

Other biomarkers such as copeptin, angiopoietin 2, and lipocalin 2 are also being investigated.

### 5.4. Invasive Hemodynamic Measurement

Although not routinely performed, invasive measurement of filling pressures can provide valuable insight into the cause of renal dysfunction in cirrhotic patients and distinguish between cardiorenal and hepatorenal syndrome. Pelayo et al. performed right heart catheterization on 127 cirrhotic patients with worsening kidney function admitted for liver transplant evaluation who met the 2015 IAC criteria for HRS-AKI. Sixty two percent of patients had elevated wedge pressure (>15 mmHg) and were subsequently switched from volume expansion to diuresis. Renal function improved and SCr decreased from 2.0 to 1.5 mg/dL (*p* = 0.003), indicating the potential significant role of accurate intravascular volume assessment using invasive hemodynamic monitoring in guiding therapy in AKI in cirrhotic patients [38].

### 5.5. Noninvasive Hemodynamic Measurement

Invasive hemodynamic monitoring cannot be performed in all cirrhotic patients with AKI. Bedside echocardiogram was utilized to assess intravascular volume status and direct management in cirrhotic patients with AKI. Premkumar et al. validated the use of inferior vena cava diameter (IVCD) and collapsibility index (IVCCI) for intravascular volume assessment in cirrhotics by correlating them with central venous pressure (CVP) measurements by right heart catheterization. The CVP value had a strong positive correlation with IVCDmax (r = 0.671, *p* = 0.037) and a clear negative linear correlation with IVCCI (r = −0.827, *p* = 0.023) [39]. Similarly, Velez et al. showed that 23% of patients had improvement in their renal function following a point-of-care echocardiography-guided therapeutic maneuver such as volume expansion, diuresis, or paracentesis [40]. VExUS (venous excess ultrasound) detects flow abnormalities in the hepatic portal vein and kidney parenchymal vein and can assess renal venous congestion in real time as well as the effect of decongestant therapy [41]. Point-of-Care Ultrasound (POCUS) has recently emerged as an important non-invasive diagnostic tool to assess intravascular volume status in cirrhotic patients with AKI. POCUS has gained increasing popularity due to ease of use and its ability to perform multiple measurements in the same setting. POCUS can accurately measure IVC diameter and percent IVC collapsibility, which are surrogates for right atrial pressure measurement and therefore can differentiate between volume deplete and volume replete states (Table 5). In one study, POCUS has been proven to better characterize intravascular volume, and prevented the misclassification of AKI in 53 cirrhotic patients [40]. Other authors have advocated for multi-organ US, including a combined ultrasonographic assessment of lung, internal jugular vein, left ventricular stroke volume, right ventricular size, superior vena cava, and/or femoral vein doppler [42].

## 6. HRS-AKI (Previously HRS-1)

### 6.1. Pathophysiology

The development of HRS occurs in the context of a systemic inflammatory response and severe hemodynamic disturbances, including splanchnic vasodilatation, peripheral arterial vasodilation, intense renal vasoconstriction, failure of renal autoregulation, cardiac dysfunction, adrenal insufficiency, and intra-abdominal hypertension.

In the early stages of cirrhosis, increased resistance to intra-hepatic blood flow leads to increased portal pressure. The resulting nitric oxide production leads to splanchnic and systemic vasodilation, with a subsequent decrease in EABV. As a result of EABV contraction, the sympathetic nervous system (SNS), renin-angiotensin-aldosterone system (RAAS), and nonosmotic vasopressin release are activated. Thus, the circulation becomes hyperdynamic with an increased heart rate, cardiac output, decreased systemic vascular resistance (SVR), hypotension, and significant renal vasoconstriction. As the architectural liver disturbances become more pronounced in the advanced stages of cirrhosis, there is further splanchnic blood pooling, systemic vasodilation, drop in blood pressure (BP), and worsening renal vasoconstriction. In end-stage liver disease (ESLD), splanchnic congestion and dilation lead to compensatory vasoconstrictor mechanisms and reduction of blood flow in extra-splanchnic beds such as the brain, lower extremities, and kidney. These pathophysiological processes lead to renal sodium retention, decreased capacity to excrete solute-free water, hyponatremia, and a drop in the GFR [43,44]. It is postulated that the release of vasodilatory substances does not affect the renal vascular bed due to the local release of counteracting vasoconstrictors such as endothelin [45].

Another theory explaining the development of HRS-AKI emphasizes the role of systemic inflammation in the pathogenesis of renal dysfunction. Systemic inflammation begins with bacterial translocation or overt bacterial infection. Bacterial pathogen-associated molecular patterns (PAMPs) activate monocytes and trigger the release of reactive oxygen species and pro-inflammatory cytokines such as tumor necrosis factor alpha (TNF-a), interleukin 6 (IL-6), and interleukin 1 beta (IL-1b). These cytokines have been implicated in AKI in decompensated cirrhosis, and contribute further to the splanchnic and pulmonary arterial vasodilation and impaired cardiac output. The end result is multi-organ dysfunction such as hepatic encephalopathy, hepato-pulmonary syndrome, renal dysfunction, and adrenal insufficiency [46]. This theory potentially explains the lack of response to vasoconstrictors in certain individuals.

### 6.2. Risk Factors

Typically, there is a specific trigger that unleashes the cascade of events leading to HRS-AKI in a predisposed individual. Predisposing factors include higher baseline creatinine, lower baseline MAP, and underlying cardiac dysfunction. Patients with cirrhotic cardiomyopathy have persistent hyperdynamic circulation which eventually leads to a diminished cardiac reserve and progressive renal dysfunction [47]. These patients are more likely to develop HRS and less likely to have AKI reversal. Common triggers of HRS-AKI are bacterial infection (mainly spontaneous bacterial peritonitis), large volume paracentesis, GI bleeding, or acute alcoholic hepatitis. These events decrease the EABV and contribute to renal vasoconstriction, which is the hallmark feature of HRS.

### 6.3. Histopathological Changes in HRS-AKI

It has long been perceived that HRS-AKI is purely a functional disorder of the kidney and does not involve any structural renal damage. These claims are supported by the reversibility of renal dysfunction with liver transplant alone [48] and by classic images showing reversibility of extreme renal vasoconstriction post-mortem [49]. Indeed, kidneys harvested from HRA-AKI patients have even been successfully transplanted [50]. However, recent evidence suggests that there is often a discrepancy between the clinical presentation and histological diagnosis. For example, Trawale et al. showed that out of 18 patients with a clinical picture compatible with HRS (SCr >1.5 mg/dL, no hematuria, and proteinuria <500 mg/day), 13 had chronic tubulo-interstitial injury, 12 had acute tubulo-interstitial injury, 10 had glomerular lesions, and 12 had vascular disease [51]. Furthermore, sustained ischemic injury from renal vasoconstriction and nephrotoxic injury due to bilirubin casts can cause intrinsic tubular injury [20]. This explains the low likelihood of renal recovery with a prolonged duration of AKI behind the recommendation for simultaneous liver and kidney transplant for patients who have been dialysis-dependent for ≥6 weeks [52]. This also explains the decreased responsiveness to pressors with increased bilirubin levels [53]. Therefore, it seems plausible that HRS may initially start as purely functional, as previously anticipated, but over time progresses to irreversible structural renal damage.

### 6.4. Prevention

HRS-AKI is easier prevented than treated. In all cirrhotic patients, norfloxacin spontaneous bacterial peritonitis (SBP) prophylaxis has been used in select candidates with cirrhosis, as it decreases the 1-year probability of SBP (7% vs. 61%) and HRS (28% vs. 41%), and improves 1-year survival (60% versus 48%) compared to placebo [54]. It is important to keep in mind, however, that quinolones can cause kidney injury themselves, mainly through acute interstitial nephritis (AIN). In patients with SBP, the incidence of HRS-AKI is lower with the concomitant sue of antibiotics and albumin compared to antibiotics alone [55]. The recommended dosage is 1.5 g/kg of body weight on day 1 and 1 g/kg of body weight on day 3, up to a maximum of 150 g/day and 100 g/day, respectively [54]. In a meta-analysis of four randomized controlled trials, the administration of albumin with antibiotics reduced patient mortality from 35.4% to 16% and reduced the incidence of renal impairment from 30.6% to 8.3% [55]. NSAIDs should be avoided in all cirrhotic patients, especially those with late-stage disease, as NSAID use is associated with HRS-AKI development. Finally, in HCV-naïve patients receiving HCV-infected organs, timely access to direct-acting antiviral drugs is essential in preventing the development of acute HCV-induced glomerulonephritis [56].

## 7. Management of AKI in Cirrhosis

### 7.1. Supportive Care

In the management of renal dysfunction in cirrhotic patients, dietary advice plays an important role. General dietary recommendations consist of sodium restriction for patients with volume overload, and the cessation of alcohol consumption. Moderate protein intake should be encouraged to avoid malnutrition and sarcopenia, but actual protein intake should be individualized in each patient according to needs, comorbidities, presence of hepatic encephalopathy, and kidney function. A thorough history should be taken to assess for the use of NSAIDs or other nephrotoxic meds that can adversely affect renal function through many mechanisms. Holding diuretics is crucial in preventing further intravascular volume contraction and hypotension. Also, non-selective beta-blockers and RAAS inhibitors should be stopped due to their BP-lowering effects. Patients are usually hospitalized and undergo a thorough search for possible underlying triggers [57]. Treating any possible infection, especially SBP, is a critical initial step of conservative management. Routine tests that should be performed include chest imaging, abdominal paracentesis, blood and urine cultures, as well as cultures of ascitic fluid. Empiric antibiotic therapy should be offered if suspicion of infection is high.

More specific management is directed at the underlying etiology. For example, patients without overt volume overload should receive judicious volume resuscitation with crystalloids or preferably with albumin challenge. For patients with suspected ATN, supportive care is offered, and paracentesis or diuretics are offered in cases where renal vein congestion and intra-abdominal hypertension are thought to be the culprit. Steroids are used in AIN, whereas cases of viral-driven glomerulonephritis benefit from treatment with direct-acting antiviral drugs. In cases of HRS-AKI, the management is often complex and necessitates multiple interventions. Supportive, pharmacological, and non-pharmacological treatments serve as a bridge towards definitive therapy which is liver transplantation in eligible patients. The goal of therapy is to reverse the hemodynamic alterations leading to renal vasoconstriction. Figure 1 demonstrates the proposed algorithm for the initial management of AKI in cirrhosis.

### 7.2. Pharmacological Management of HRS-AKI

The mainstay of pharmacological therapy in HRS-AKI consists of plasma expanders and vasoconstrictors. The combination of albumin and vasopressors is a treatment of choice for HRS-AKI, according to the American association for the study of liver disease (AASLD) and European society for liver disease [12,58]. Many factors can affect response to vasoconstrictors such as timing of initiation after diagnosis, degree of renal dysfunction, degree of cholestasis, post-vasoconstrictor increase in mean arterial pressure (MAP), reversal of underlying triggers, and the presence of underlying cirrhotic cardiomyopathy or portopulmonary hypertension [59,60,61].

#### 7.2.1. Albumin

A lack of improvement in renal function following an albumin challenge of 1 g/kg of body weight per day for 2 days is currently a requirement for a diagnosis of HRS-AKI. Beyond its role as a plasma expander related to its oncotic properties, human albumin likely exerts many effects in decompensated cirrhosis that target the underlying pathophysiological mechanisms leading to HRS [62]. First, albumin is thought to decrease the burden of systemic inflammation and oxidative stress by acting as a free radical scavenger and metal ion chelator. It plays a role in immunomodulation by binding to endotoxins and decreasing TNF-induced nuclear factor-kappa B activation. It also helps stabilize the endothelium thus maintaining capillary integrity and decreasing permeability. Other effects include: (1) positive cardiac inotropy, (2) anti-coagulant and anti-thrombotic agent, and (3) binding and transport of drugs, ions, bile acids, and bilirubin [63,64]. When 777 patients with decompensated liver cirrhosis in the United Kingdom were randomized to receive an albumin infusion to target a serum albumin of ≥30 g/L versus standard of care, albumin did not provide additional benefit in terms of lower rates of renal dysfunction, infection, or death between days 3 and 15 after the initiation of treatment. Furthermore, the albumin group experienced an increased risk of serious and life-threatening adverse events, including pulmonary edema [65]. However, in the case of HRS-AKI, albumin alone was shown to reverse HRS in 15% of patients [66]. Salerno et al. performed a meta-analysis of 19 clinical studies with 574 patients in total, and found that each 100 g increase in cumulative albumin dose was associated with increased survival in patients with HRS-AKI (HR 1.15; 95% CI 1.02–1.31; *p* = 0.023) [67]. Perhaps the most robust evidence regarding human albumin use in cirrhosis comes from a recent overview of 300 papers, including 18 meta-analyses performed by Qi et al. The group concluded that albumin reduced mortality and renal impairment exclusively in those with SBP but not in those with non-SBP infections [68]. Despite its widespread use in AKI in cirrhotic patients, the optimal albumin dose is uncertain, and one should be particularly vigilant about the possibility of iatrogenic pulmonary edema with the use of higher doses. Additional studies are needed to guide albumin replacement and tailor dosing to each individual patient’s needs. Albumin and terlipressin are currently recommended as first-line therapy for HRS-AKI by the European Association for the Study of the Liver Practice Guidelines [58].

#### 7.2.2. Octreotide and Midodrine

Midodrine is an oral selective alpha 1 agonist which works primarily by counteracting systemic vasodilation, and octreotide is a somatostatin analog which is usually administered subcutaneously or intravenously. Both agents are usually administered in combination with albumin. Although the use of this combination is supported by some level of evidence [69,70,71], a randomized controlled trial has clearly shown that midodrine/octreotide are less effective than norepinephrine in achieving renal response and improving 30-day survival [72]. Midodrine, octreotide, and albumin combination is also inferior to a combination of terlipressin and albumin in terms of renal recovery from HRS-AKI (28.6% vs. 70.4%) [73].

#### 7.2.3. Norepinephrine

Norepinephrine is an intravenous alpha 1 adrenergic agonist used for the treatment of HRS-AKI and is a reasonable alternative in countries where terlipressin is not available. It induces splanchnic vasoconstriction with limited effect on the myocardium. The evidence for epinephrine use comes from small studies. A meta-analysis by Nassar Junior et al. included 154 patients from four studies comparing norepinephrine and terlipressin. No difference was found between the two pressors in the reversal of HRS (RR = 0.97, 95% CI = 0.76 to 1.23), mortality at 30 days (RR = 0.89, 95% CI = 0.68 to 1.17), or recurrence of HRS (RR = 0.72; 95% CI = 0.36 to 1.45). Adverse events were less common with norepinephrine (RR = 0.36, 95% CI = 0.15 to 0.83). However, all the included studies were deemed to be at high risk of bias [74]. Gupta et al. studied 30 patients with HRS type 1 (now HRS-AKI) and found that norepinephrine plus albumin administered for 14 days was associated with a 73% response rate (as evidenced by a decrease in SCr to <1.5 mg/dL, and increase in creatinine clearance, urine output, MAP, and serum sodium) [75]. One of the major drawbacks to norepinephrine use has been the need for monitoring in the intensive care unit, thus mobilizing significant resources. To address this major limitation, Kwong et al. conducted a pragmatic study where 20 patients with HRS who had not responded to midodrine and octreotide were administered norepinephrine in a non-ICU setting. The starting dose was 5 µg/minute, with protocolized progressive increments overseen by a hepatologist to achieve a MAP of 10 mm Hg above baseline. Forty five percent of these patients had a full or partial response to norepinephrine [76]. These results suggest that norepinephrine use outside of the ICU may be safe and effective if performed in a controlled environment and patients are chosen carefully.

#### 7.2.4. Terlipressin

In September 2022, terlipressin received FDA approval for the treatment of adults with HRS-AKI, making it the first approved treatment for this indication in the United States [77]. However, terlipressin has been used widely outside the US for over 30 years and is considered standard of care in many centers in Europe and around the world [78]. Terlipressin is a 12-amino acid peptide that is derived from lysine vasopressin and works as a vasopressin receptor agonist with increased selectivity for V1 and V2 receptors [79]. Terlipressin lowers portal blood flow and portal pressure by inducing splanchnic vasoconstriction and shunting splanchnic blood back to the systemic circulation, which leads to an increase in EABV with subsequent improvement in mean arterial pressure and renal perfusion, thereby reversing the pathophysiology associated with HRS.

Three major randomized controlled trials conducted in North America showed the benefit of terlipressin in reversing HRS, reducing ICU stay, and improving renal replacement therapy: OT-0401, REVERSE, and CONFIRM [80,81]. The recent FDA approval of terlipressin was based on the results of the CONFIRM trial [82]. In this randomized, double, blind, placebo-controlled phase 3 trial, 300 patients with HRS-1 were randomly assigned to receive terlipressin (199 participants) or placebo (101 participants) for a maximum of 14 days. Albumin was used in both arms. The primary endpoint was HRS reversal as defined by 2 days of SCr of 1.5 mg/dL or less, obtained at least two hours apart, by day 14 or by the participant’s final day in the study. Thirty two percent of patients on terlipressin were able to achieve HRS reversal, vs. 17% in the placebo arm (*p* = 0.006). A greater percentage of patients in the terlipressin group had a SCr value of 1.5 mg/dL or less while on treatment at day 14 or at discharge. The need for renal replacement therapy at day 30 was also lower in the terlipressin-treated patients compared to the placebo group (32% vs. 16%; *p* = 0.003). In a subgroup analysis of those with systemic inflammatory response syndrome, a greater percentage had HRS reversal in the terlipressin group vs. placebo group (84% vs. 28%; *p* <0.001).

The usual terlipressin dose is 1 mg IV every 6 h for 3 days. Dosing is then adjusted based on renal function on the fourth day of treatment: if the SCr value has decreased by ≥30% from the level at initiation of therapy, the dose of 1 mg q 6 h should be continued. If the SCr value has decreased by <30% from the level at initiation of therapy, the dose should be increased to 2 mg every 6 h and if the SCr value is equal to or above the level at initiation of therapy, treatment should be discontinued. The most commonly observed adverse effects (AEs) reported in the CONFIRM trial were abdominal pain (19.5%), nausea (16%), respiratory failure (15.5%), diarrhea (13%), and dyspnea (12.5%). Other reported AE include headache, hyponatremia, ischemic skin necrosis, and gangrene [83,84,85,86]. Given the incidence of respiratory failure observed in the CONFIRM trial, a boxed warning for serious or fatal respiratory failure was issued. Although it is not clear if this was a direct effect of terlipressin or related to increased albumin-based resuscitation in the treatment group [87], terlipressin is contraindicated in patients experiencing hypoxia or worsening respiratory symptoms and should be discontinued if these complications occur. Patients should be closely monitored for changes in respiratory status if terlipressin is administered. Terlipressin is also contraindicated in patients with myocardial or intestinal ischemia. It is therefore critically important to carefully select therapy candidates and to avoid use in patients unlikely to draw sufficient benefit. For example, if a patient has high priority for a transplant (MELD ≥35), the benefit of terlipressin is unlikely to outweigh the risk and terlipressin-related AEs could lead to ineligibility for liver transplantation. In addition, patients with SCr >5 mg/dL are unlikely to benefit from terlipressin and patients with acute-on-chronic liver failure (ACLF) Grade 3 should not receive terlipressin due to significant risk of respiratory failure [77,88,89].

### 7.3. Non-Pharmacological Management

Non-pharmacological management for HRS-AKI includes transjugular intrahepatic portosystemic shunting (TIPS), artificial liver support systems, and renal replacement therapy (RRT). These therapies are usually used as a bridge towards liver transplantation which is considered the only definitive treatment.

#### 7.3.1. Tips

TIPS can lead to HRS reversal and improvement in renal and cardiac function through a reduction in portal hypertension and SNS/RAAS activation [90,91,92] but should only be used in carefully selected candidates [93].

#### 7.3.2. Artificial Liver Support

Artificial liver support systems include the Molecular Adsorbent Recirculating System™ (MARS™), the Single-Pass Albumin Dialysis system (SPAD), and the Fractionated Plasma Separation and Adsorption system (Prometheus™) [94]. MARS allows simultaneous liver and kidney detoxification by combining the continuous renal replacement therapy (CRRT) technique with albumin-enriched dialysate. MARS removes both water-soluble and albumin-bound molecules such as bilirubin, ammonium, urea, creatinine, fatty acid, bile salt, and inflammatory cytokines including TNFα and IL6. Evidence supporting its utility in HRS is conflicting and the largest randomized controlled trial investigating its role did not find a significant difference in the overall 6-month, 6-month transplant-free, and 1-year survivals [95].

#### 7.3.3. Renal Replacement Therapy

Renal replacement therapy (RRT) in patients with decompensated liver failure and AKI is controversial and its practice is inconsistent around the world. It is well known that RRT is particularly challenging in patients with cirrhosis, secondary to the increased risk of secondary complications such as bleeding from vascular access, intradialytic hypotension, and increased risk of cardiac events. Splanchnic vasodilation results in decreased effective arterial blood volume and difficulty with volume management, especially on intermittent hemodialysis. CRRT is more hemodynamically tolerated but necessitates an ICU level of care. In addition, citrate anticoagulation is poorly tolerated in patients with liver failure and frequently results in citrate toxicity. The prognosis of patients with HRS-AKI requiring RRT is unfortunately poor. In fact, each day of CRRT is associated with an increased OR of death of 1.39 (95% CI 1.01–1.90; *p* = 0.04) [96]. Allegretti et al. found that mortality rates were equally high whether the AKI was caused by ATN or HRS (median transplant-free survival: 15 days for HRS vs. 14 days for ATN, *p* = 0.60) [97]. Another study from Europe conducted by Staufer et al. in 193 patients with AKI and cirrhosis showed that the group that initiated RRT had higher 28-day mortality compared to the group who did not initiate RRT (83% vs. 30%, *p* <0.001) [98]. Therefore, many feel that RRT should mainly be offered to LT candidates or to patients who are undergoing the evaluation process as long as this decision can be reversed if they become ineligible [99]. For transplant-ineligible patients, the decision should be individualized and should take into account patient preferences and family wishes while carefully weighing the risks and benefits.

#### 7.3.4. Liver Transplantation

While successful liver transplant is considered the optimum therapy for ESLD, it does not always result in improvement or normalization of renal function. Simultaneous liver and kidney transplant (SLK) improves survival compared with liver transplant alone (LTA) in patients with advanced and prolonged renal dysfunction [100], especially in those requiring dialysis [101,102]. However, the need for SLK is not always clear in all patients. Indeed, in some patients with severe HRS-AKI, improvement in kidney function can occur following LTA [103]. In fact, Brennan et al. demonstrated that up to 87% of patients with renal dysfunction who received LTA had improvement in renal function within 1 month of transplant, and that the prevalence of stage 4 or 5 CKD after 1 year was as low as 6.8% [104]. A potentially useful tool in predicting renal histology in cirrhosis and thus renal recovery is arterial blood pressure at the time of liver transplant evaluation: a systolic blood pressure (BP) of ≤90 mm Hg correlates with 100% sensitivity and 98% specificity with normal biopsies or ATN, whereas a systolic BP of ≥140 mm Hg correlates with 22% sensitivity and 90% specificity with advanced interstitial fibrosis (IF) and glomerulosclerosis [105].

In order to avoid futile kidney transplants for recipients with a low likelihood of end-stage renal disease (ESRD) post-LTA, and to contain the significant surge in the rate of SLK following the 2002 MELD score implementation [106], the Organ Procurement and Transplantation Network (OPTN) revised its SLK allocation policy in 2017 [107]. In accordance with the new UNOS policy, liver transplant candidates who meet one of the following criteria are eligible for SLK allocation: 1. CKD with a GFR of ≤60 mL/min for >90 days and ESRD on dialysis or a GFR ≤30 mL/min at the time of waiting list registration, 2. Sustained AKI for 6 weeks with dialysis requirement at least once every seven days or a GFR of ≤25 mL/min at least once every seven days, and 3. Metabolic disease such as hyperoxaluria, atypical hemolytic uremic syndrome from mutations in factor H or factor I, familial non-neuropathic systemic amyloidosis, or methylmalonic aciduria. Beside the establishment of formal SLK eligibility criteria, the concept of a “safety-net” was introduced for the first time. As a result of this policy change, liver transplant recipients who are either on dialysis or have an eGFR of ≤20 mL/min between 60 and 365 days after liver transplant receive priority on the waiting list for deceased kidney donors with a KDPI between 20% and 85%.

## 8. Conclusions

Our understanding of the classification, diagnosis, and pathophysiology of AKI in cirrhosis has made great strides over the last 10 years. Effective therapies are now available to reverse the underlying mechanism of renal dysfunction and avoid progression of renal failure. However, there is still a critical unmet need for studies that guide albumin and vasopressor dosing and predict reversibility of renal failure after liver transplant. Further studies are also needed to validate biomarkers and point-of-care ultrasonography and bring them into day-to-day clinical use.

## Figures and Tables

**Figure 1 diagnostics-13-02361-f001:**
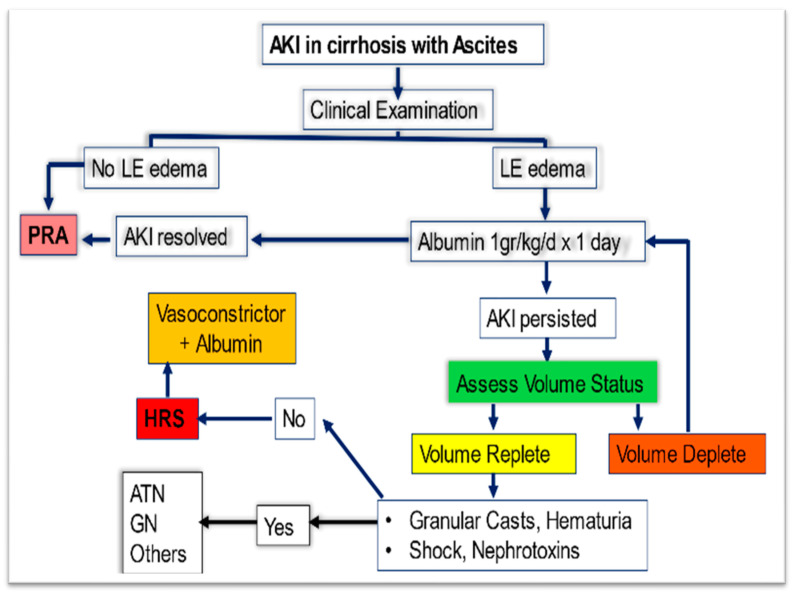
Suggested algorithm for the initial management of AKI in patients with cirrhosis and ascites. Careful clinical examination and frequent assessment of intravascular volume status by measuring inferior vena cava (IVC) diameter and collapsibility using Point-of-Care Ultrasound (POCUS) or bedside echocardiogram are needed to avoid volume overload from overzealous albumin infusion.

**Table 1 diagnostics-13-02361-t001:** New definition and staging system of HRS in liver cirrhosis.

Old Classification	New Classification	Diagnostic Criteria		Stages
**HRS-1**	**HRS-AKI**	a. Absolute increase in sCr ≥0.3 mg/dL within 48 h **and/or** b. Percent increase in sCr ≥50% using the last available value of outpatient sCr within 3 months as the baseline value		**Stage 1:** increase in SCr ≥0.3 mg/dL (26.5 μmol/L) within 48 h or increase ≥1.5–1.9 fold from baseline 1a: SCr <1.5 mg/dL 1b: SCr >1.5 mg/dL
				**Stage 2:** increase in SCr 2–3 fold from baseline
				**Stage 3:** increase in SCr ≥3-fold from baseline or SCr ≥4.0 mg/dL (353.6 μmol/L) with an acute increase ≥0.3 mg/dL (26.5 μmol/L) or initiation of renal-replacement therapy
**HRS-2**	**HRS-NAKI**	**HRS-AKD**	eGFR < 60 mL/min per 1.73 m^2^ for <3 months in the absence of other causes	
		**HRS-CKD**	eGFR < 60 mL/min per 1.73 m^2^ for ≥3 months in the absence of other causes	

**Table 2 diagnostics-13-02361-t002:** Differential diagnoses for AKI in cirrhosis and potential causes, diagnosis, and management.

Etiology	Potential Causes and Pathophysiology	Diagnosis	Management
**Pre-renal AKI**	Decreased oral intake, use of diuretics for ascites, use of laxatives for hepatic encephalopathy prophylaxis	Clinical history, POCUS findings, bland urinary sediment	Discontinuation of diuretics and repletion of intravascular volume preferably with albumin.
**Ischemic acute tubular necrosis (ATN)**	Prolonged pre-renal insult, gastrointestinal bleed leading to hypovolemic shock, septic shock due to spontaneous bacterial peritonitis (SBP)	Clinical history, granular casts on urine microscopy	Conservative, diuretics for volume overload as needed, renal replacement therapy (RRT)
**Toxic ATN**	Nephrotoxic medications such as vancomycin or fluoroquinolones used for SBP treatment	Clinical history, granular casts on urine microscopy	Conservative, diuretics for volume overload as needed, RRT
**Bile cast nephropathy (aka cholemic nephropathy)**	Deposition of intra-tubular bilirubin casts in severe liver failure	Serum bilirubin levels typically >10 mg/dL, bilirubin casts on urine microscopy	Liver transplant to decrease serum bilirubin levels, diuretics for volume overload as needed, RRT
**HRS-AKI (formerly HRS-1)**	Splanchnic vasodilatation, peripheral arterial vasodilation, and intense renal vasoconstriction	Diagnosis of exclusion in the absence of shock, nephrotoxic drug exposure, or structural kidney disease (proteinuria >500 mg per day, microhematuria >50 red blood cells per high-power field, and/or abnormal renal ultrasonography)	Albumin, vasopressors (norepinephrine, terlipressin)
**Cirrhotic cardiomyopathy**	Hyperdynamic circulation leading to renin-angiotensin-aldosterone system (RAAS) and sympathetic nervous system (SNS) activation, cardiosuppressants such as nitric oxide and inflammatory cytokines	Echocardiography	Liver transplant, diuretics to help reduce preload, vasopressors
**Abdominal compartment syndrome**	Tense ascites causing severe intra-abdominal hypertension and renal vein congestion	Sustained intra-abdominal pressure >20 mmHg	Large-volume paracentesis
**Secondary immunoglobulin A (IgA) nephropathy**	Decrease in the expression of the hepatic sialo-glycoprotein receptor leading to defective IgA glycosylation	Hematuria and proteinuria on urinalysis, renal biopsy	Liver transplant
**Membranoproliferative glomerulonephritis (MPGN)**	Hepatitis C virus (HCV) (frequently leads to cryoglobulinemic vasculitis), or Hepatitis B virus (HBV)	Hematuria and proteinuria on urinalysis, red blood cell casts on urine microscopy, positive HCV RNA or HBV DNA by polymerase chain reaction (PCR)	Direct-acting antiviral drugs
**Acute Interstitial nephritis (AIN)**	Fluoroquinolone use for SBP prophylaxis or proton-pump inhibitor (PPI) use for GI prophylaxis	Clinical history, sterile pyuria on urinalysis, white blood cell casts on urine microscopy	Withdrawal of offending agent
**Obstructive uropathy**	Midodrine (alpha agonist used for blood pressure support in HRS)	Physical exam, bladder scan, POCUS, renal ultrasonography	Withdrawal of offending medication, urinary catheterization

**Table 3 diagnostics-13-02361-t003:** Differences between the major 3 causes of AKI in cirrhotic patients.

	PRA	ATN	HRS
**Hypotension**	Yes	Yes	Yes
**Shock**	No	**Yes**	No
**Nephrotoxins**	No	**Yes**	No
**Ascites**	+/−	+/−	**+**
**Response to IV Albumin**	**Yes**	No	No
**IVC**	**<2.5 cm, >50% collapse**	>2.5 cm, <50% collapse	>2.5 cm, <50% collapse
**Urine sediment**	Negative	**Granular casts**	Negative
**FeNa**	<1%	<1%	<1% (<0.1%)
**Urinary Na**	<10 mEq/L	<10> mEq/L	<10 mEq/L
**Urine Biomarkers (NGAL)**	+	**+++**	+

**Table 4 diagnostics-13-02361-t004:** Novel biomarkers for the diagnosis of AKI in cirrhosis.

Biomarker	Description	Utility	Pitfalls
**Cystatin C**	- Cysteine protease inhibitor - Produced by all nucleated cells in the body - Filtered by the glomerulus and metabolized in the tubules	- Less influenced by muscle mass, age, and diabetes than serum creatinine [26] - Useful in AKI prediction and prognostication: MELD-Cystatin C improves predictive accuracy of mortality [27] - Equations based on both creatinine and Cysatin C least biased in assessing the GFR in cirrhosis [28]	- Several non-GFR determinants of higher Cystatin C such as male sex, greater height and weight, higher lean body mass, higher fat mass, diabetes, higher levels of inflammatory markers, hyper- and hypothyroidism, and glucocorticoid use [29,30]
**Urinary NGAL**	- Produced by neutrophils and epithelial cells including kidney tubular cells - Abundantly expressed in the urine following ischemic injury	- Marker of tubular damage - Useful for differentiating pre-renal AKI from ATN (highest in ATN) - Greatest accuracy among monomeric NGAL (mNGAL), interleukin (IL)-18, and other conventional urinary biomarkers for differential diagnosis between ATN and other types of AKI when measured at day 3 in decompensated cirrhosis [31] - Predictor of 90-day patient transplant-free survival [32,33] and prognostic factor for mortality in ACLF [34] - Starts to rise after 3 h in the urine following renal injury [35]	- Lack of standardization - Uncertainty regarding the cutoff value - Unavailable in many countries thus making it a research only test
**Urinary KIM-1**	- Transmembrane protein that is upregulated in the proximal tubule and shed in the urine in response to ischemia	- Rises 2–3 h following kidney injury - Useful in differentiating types of AKI and predicting patient mortality - Highest in ATN	- Lack of standardization - Poor sensitivity and specificity [36]
**Urinary L-FABP**	- Intracellular lipid chaperone involved in lipid-mediated processes	- Promising prognostic biomarker in patients with decompensated cirrhosis [37] - Highest in ATN	- Limited studies in decompensated cirrhosis
**Urinary IL-18**	- Proinflammatory cytokine, expressed in the proximal tubular cells - Upregulated in acute ischemic injury	- Marker of tubular damage: higher in ATN compared with pre-renal azotemia, UIT, and CKD	- Does not predict patient mortality or kidney outcomes

**Table 5 diagnostics-13-02361-t005:** POCUS Assessment of Inferior Vena Cava (IVC) and correlation with Central Venous Pressure (CVP).

IVC Diameter (cm)	Respiratory Variation (Collapse)	CVP (cm H_2_O)
**<1.5**	**Total collapse**	**0–5**
**1.5–2.5**	**>50%**	**6–10**
**1.5–2.5**	**<50%**	**11–15**
**>2.5**	**<50%**	**16–20**
**>2.5**	**No change**	**>20**

IVC < 2.5 cm and <50% collapse can be caused by intra-abdominal hypertension.

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
