# Peer review of "Acute Kidney Injury in Liver Cirrhosis"

_diagnostics, 2023, doi:10.3390/diagnostics13142361_

Round 1

Reviewer 1 Report

Introduction

In the line 23, “prolonged hospitalization, increased prevalence of acute-on-chronic liver failure and decreased survival” cannot be attributed to the complications of AKI.

Definitions

In the line 34, the citation 10 and the content expressed are inconsistent.

In the lines 39 and 40, according to the paper “Angeli P, Gines P, Wong F, et al. Diagnosis and management of acute kidney injury in patients with cirrhosis: revised consensus recommendations of the International Club of Ascites. Gut. 2015;64(4):531-537.”, an increase in SCr of twofold from baseline belongs to stage 1 of acute kidney injury, and an increase in SCr of threefold from baseline belongs to stage 2.

In the line of 59, “SCr ≥ 1.5 mg/dL”, rather than “SCr > 1.5 mg/dL”, belongs to stage 1b of HRS-AKI.

 Some recent articles regarding HRS should be reviewed and discussed (PMID: 30474439; 36697778; 35601800).

Epidemiology and prognostic implications

In the line of 98, the patients included in the citation were “cirrhosis and renal failure” rather than “cirrhosis and AKI”, which is inconsistent with your statement.

Differentiating different causes of AKI in cirrhosis

In the line of 122, “was” should be revised as “is”.

The Table 3 is incorrectly placed.

Management of AKI in cirrhosis

In the line of 287, you should illustrate the target serum albumin higher or lower than 30g/L.

In the line of 306, you said “randomized controlled trials”, so there should be many citations instead of one.

Main body

Abbreviations and full names should be checked carefully, such as in the line 37 of “ESRD”, 91 of “ICA”, and 83 of “Hepatorenal syndrome”.

Format should be checked carefully. You should add a comma before “and”, because the “and” connects two sentences. In the line of 61, you should write “Huelin et al.” instead of “Huelin et al”. There are two “a)” to describe subheadings in the section of Management of AKI in cirrhosis. In the line of 281, there is an extra space between “metal” and “ion”. In the line of 340, the “. It” should be deleted. In the line of 365, “If” should be “if”.

Author Response

Response to Reviewer 1:

We thank the Reviewer for his thorough review of the paper.

Point #1

The Reviewer argues that AKI might not be related to increasing the prevalence of ACLF.

Response:

We agree with the Reviewer. The statement relating AKI with ACLF has been deleted.

Point # 2

The Reviewer is asking to align the paper's content with citation #10.

Response:

We agree with the Reviewer. We changed the citation. 

Point #3

The Reviewer is commenting on the paper by Angeli et al. and the SCr increase to stage AKI.

Response

We reviewed again the paper by Angeli et al. and an increase of baseline SCr by 1.5-1.9 folds belong to AKI stage 1 while a 2-3 folds increase indicates stage 2 AKI. No changes have been made.

Point #4

The Reviewer is pointing out that in the line of 59, “SCr ≥ 1.5 mg/dL”, rather than “SCr > 1.5 mg/dL”, belongs to stage 1b of HRS-AKI.

Response

We agree with the Reviewer. Changes have been made.

Point #5

The Reviewer is asking to add more references to recent articles regarding HRS

Response:

We agree with the Reviewers. We added some of the suggested references to the best of our ability.

Point #6

The Reviewer is pointing out that in line 98, the patients included in the citation were “cirrhosis and renal failure” rather than “cirrhosis and AKI”, which is inconsistent with our statement.

Response

We understand the Reviewer's point but in this particular study, 91% of the patients had AKI either because of HRS, ATN, or prerenal azotemia and only 9% had parenchymal nephropathy. We changed  the manuscript to reflect that

Changes

in a prospective study of 562 patients with renal failure and cirrhosis (91% of these patients had AKI)  published in 2011

 Point # 7

The reviewer is pointing out to typos in the English language.

Response

We agree with the Reviewer. These have been fixed

Point #8

The Reviewer is pointing out that Table 3 is misplaced

Response

We changed the title of Table 3 based on Reviewer #2 request. I hope that this has addressed Reviewer #1 concern.

Point #9

The Reviewer is asking to clarify the target serum albumin used in this study.

Response

We agree with the Reviewer. The statement was changed to reflect that the target serum albumin was >=30 gr/L

Point #10

The Reviewer is asking for more than one citation regarding midodrine and octreotide

Response

More than one citation has been provided in the manuscript including reference 74 and reference 75.

Point #11

The Reviewer is asking to clarify abbreviations

Response

Appropriate changes have been made.

 Point #12

The Reviewer is asking to revise some formatting concerns

Response

Appropriate changes have been made.

Reviewer 2 Report

Dear authors, I want to congratulate you on this review. It is an important topic for the hepatologist, but also for the nephrologist. The text needs some corrections, as listed below:

- line 9 - in abstract you repeated some words ..." with worse survival with the worst survival…"

- line 151-153 - this sentence is unclear: “Premkumar et al. validated the use of inferior vena cava diameter (IVCD) and collapsibility index (IVCCI) for intravascular volume assessment in cirrhotics by showing their positive correlation between central venous pressure (CVP)” – maybe something is missing?

- please explain all the abbreviations from text – ex. SBP

- please, move table 3 title before table - Table 3: key diagnostic features of PRA, ATN and HRS

I would have a few more suggestions:

-          The Introduction section can be improved

-          Management of renal dysfunction in cirrhotic patients I believe must begin with diet. It is very important to know what you recommend your patient to eat/drink. I'm a pediatric nephrologist and maybe this is a bit easy, but for adults it's very important to limit protein to avoid elevated ammonia and risk portal encephalopathy. It is also important to limit alcohol and nephrotoxic and hepatotoxic drugs.

-          Therapy with holding diuretics, NSAIDs, non-selective beta blockers, RAAS inhibitors should perhaps have been developed, as well as the non-pharmacological methods, briefly described.

-          I would be very cautious in the use of first-generation quinolones in the management of HRS-associated infections, due to the risk of additional renal tubular damage.

Good luck!

Minor editing of English language required

Author Response

Response to Reviewer #2

We would like to thank this Reviewer for his/her encouraging remarks regarding our review article.

Point #1

The Reviewer is asking to avoid repeating the words “worse” and “worst” words in the abstract section.

Response

We agree with the Reviewer. The abstract has been changed to reflect this change.

Point #2

The Reviewer is asking to further clarify the sentence discussing the relationship between IVC collapsibility and CVP

Response

We agree with the Reviewer. This statement has been changed to the following Premkumar et al. validated the use of inferior vena cava diameter (IVCD) and collapsibility index (IVCCI) for intravascular volume assessment in cirrhotics by correlating them with central venous pressure (CVP) measurements obtained via right heart catheterization. CVP had a strong positive correlation with IVCDmax (r = 0.671, P = 0.037) and a clear negative linear correlation with IVCCI (r = −0.827, P = 0.023) (27).  

Point #3

The Reviewer is asking to clarify all the abbreviations

Response

We agree with the Reviewer. Necessary changes have been made.

Point#4

The Reviewer is asking to move the Title of Table 3.

Response

We agree with the Reviewer. The title has been moved and a more appropriate title has been included.

Point #5

The Reviewer is asking to improve the introduction section

Response

We agree with the Reviewer but due to space constraints and the fact that we had a very extensive review no changes were undertaken.

Point #6, Point #7 and Point #8

The Reviewer is asking to add a section regarding general dietary and medication management in patients with AKI and cirrhosis as well as caution with first-generation quinolone use

Response

We thank the Reviewer for this excellent recommendation. A paragraph highlighting general dietary and medication changes as well as caution with quinolones has been included

Changes

In the management of renal dysfunction in cirrhotic patients, dietary advice plays an important role. General dietary recommendations consist of sodium restriction for patients with volume overload and cessation of alcohol consumption. Moderate protein intake should be encouraged to avoid malnutrition and sarcopenia but actual protein intake should be individualized in each patient according to needs, comorbidities, presence of hepatic encephalopathy and kidney function. A thorough history should be taken to assess for the use of NSAIDs or other nephrotoxic meds that can adversely affect renal function through many mechanisms. Holding diuretics is crucial in preventing further intravascular volume contraction and hypotension. Also, non-selective beta-blockers and RAAS inhibitors are stopped due to their BP-lowering effects. Patients are usually hospitalized and undergo a thorough search for possible underlying triggers (45).  Treating any possible infection, especially SBP, is a critical initial step of conservative management.  Routine tests that should be performed include chest imaging, abdominal paracentesis, blood and urine cultures as well as cultures of ascitic fluid. Empiric antibiotic therapy should be offered if suspicion of infection is high.

Round 2

Reviewer 1 Report

No further comment

No further comment

Reviewer 2 Report

Dear authors,

I want to congratulate you on this review. 

Thank you for taking the recommendations into account and modifying the article accordingly to obtain a better form for publication.

I have no other comments.

Best regards